# HOMOGENEOUS LINEAR INEQUALITY CONSTRAINTS FOR NEURAL NETWORK ACTIVATIONS

## ABSTRACT

We propose a method to impose homogeneous linear inequality constraints of the form $Ax \leq 0$ on neural network activations. The proposed method allows a data-driven training approach to be combined with modeling prior knowledge about the task. One way to achieve this task is by means of a projection step at test time after unconstrained training. However, this is an expensive operation. By directly incorporating the constraints into the architecture, we can significantly speed-up inference at test time; for instance, our experiments show a speed-up of up to two orders of magnitude over a projection method. Our algorithm computes a suitable parameterization of the feasible set at initialization and uses standard variants of stochastic gradient descent to find solutions to the constrained network. Thus, the modeling constraints are always satisfied during training. Crucially, our approach avoids to solve an optimization problem at each training step or to manually trade-off data and constraint fidelity with additional hyperparameters. We consider constrained generative modeling as an important application domain and experimentally demonstrate the proposed method by constraining a variational autoencoder.

## 1 INTRODUCTION

Deep learning models (LeCun et al., 2015) have demonstrated remarkable success in tasks that require exploitation of subtle correlations, such as computer vision (Krizhevsky et al., 2012) and sequence learning (Sutskever et al., 2014). Typically, humans have strong prior knowledge about a task, e.g., based on symmetry, geometry, or physics. Learning such a priori assumptions in a purely data-driven manner is inefficient and, in some situations, may not be feasible at all. While certain prior knowledge was successfully imposed – for example translational symmetry through convolutional architectures (LeCun et al., 1998) – incorporating more general modeling assumptions in the training of deep networks remains an open challenge. Recently, generative neural networks have advanced significantly (Goodfellow et al., 2014; Kingma & Welling, 2014). With such models, controlling the generative process beyond a data-driven, black-box approach is particularly important.

In this paper, we present a method to impose prior knowledge through homogeneous linear inequality constraints of the form $Ax \leq 0$ on the activations of deep learning models. We directly impose these constraints through a suitable parameterization of the feasible set. This has several advantages:

- The constraints are hard-constraints in the sense that they are satisfied at any point during training and inference.
- Inference on the constrained network incurs no overhead compared to unconstrained inference.
- There is no manual trade-off between constraint satisfaction and data representation.

In summary, the main contribution of our method is a reparameterization that incorporates homogeneous linear inequality hard-constraints on neural network activations and allows for efficient test time predictions, i.e., our method is faster up to two orders of magnitude. The model can be optimized by standard variants of stochastic gradient descent. As an application in generative modeling, we demonstrate that our method is able to produce authentic samples from a variational autoencoder while satisfying the imposed constraints.

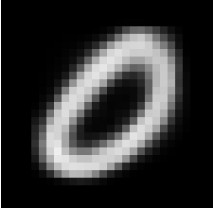 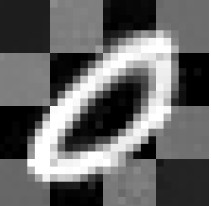

Figure 1: Samples drawn from a variational autoencoder trained on MNIST without constraints (left) and with a checkerboard constraint on the output domain (right). For a pixel intensity domain $[-1, 1]$, the checkerboard constraint forces the image tiles to have average positive or negative brightness.

## 2 RELATED WORK

Various works have introduced methods to impose some type of hard constraint on neural network activations. This differs from a classical constrained optimization problem (Nocedal & Wright, 2006) in that the constraints are on the image of a parameterized function rather than on the neural network parameters.

Márquez-Neila et al. (2017) formulated generic differentiable equality constraints as soft constraints and employed a Lagrangian approach to train their model. While this is a principled approach to constrained optimization, it does not scale well to practical deep neural network models with their vast number of parameters. To make their method computationally tractable, a subset of the constraints is selected at each training step. In addition, these constraints are locally linearized; thus, there is no guarantee that this subset will be satisfied after a parameter update.

For the specific problem of weakly supervised segmentation, Pathak et al. (2015) proposed an optimization scheme that alternates between optimizing the deep learning model and fitting a constrained distribution to these intermediate models. However, this method involves solving a (convex) optimization problem at each training step. Furthermore, the overall convergence path depends on how the alternating optimization steps are combined, which introduces an additional hyperparameter that must be tuned. Briq et al. (2018) approached the weakly supervised segmentation problem with a layer that implements the orthogonal projection onto a simplex, thereby directly constraining the activations to a probability distribution. This optimization problem can be solved efficiently, but does not generalize to other types of inequality constraints.

OptNet, an approach to solve a generic quadratic program as a differentiable network layer, was proposed by Amos & Kolter (2017). OptNet backpropagates through the first-order optimality conditions of the quadratic program, and linear inequality constraints can be enforced as a special case. The formulation is flexible; however, it scales cubically with the number of variables and constraints. Thus, it becomes prohibitively expensive to train large-scale deep learning models.

Finally, several works have proposed handcrafted solutions for specific applications, such as skeleton prediction (Zhou et al., 2016) and prediction of rigid body motion (Byravan & Fox, 2017). In contrast, to avoid laborious architecture design, we argue for the value of generically modeling constraint classes. In practice, this makes constraint methods more accessible for a broader class of problems.

**Contribution**    In this work, we tackle the problem of imposing homogeneous linear inequality constraints on neural network activations. Rather than solving an optimization problem during training, we split this task into a *feasibility step* at initialization and an *optimality step* during training. At initialization, we compute a suitable parameterization of the constraint set (a polyhedral cone) and use the neural network training algorithm to find a good solution within this feasible set. Conceptually, we are trading-off computational cost during initialization to obtain a model that has no overhead at test time. The proposed method is implemented as a neural network layer that is specified by a set of homogeneous linear inequalities and whose output parameterizes the feasible set.

## 3 Linear inequality constraints for deep learning models

We consider a generic $L$ layer neural network $F_\theta$ with model parameters $\theta$ for inputs $x$ as follows:

$$F_\theta(x) = f_{\theta_L}^{(L)}(\sigma(f_{\theta_{L-1}}^{(L-1)}(\sigma(\ldots f_{\theta_1}^{(1)}(x)\ldots)))), \tag{1}$$

where $f_{\theta_l}^{(l)}$ are affine functions, e.g., a fully-connected or convolutional layer, and $\sigma$ is an element-wise non-linearity[1], e.g., a sigmoid or rectified linear unit (ReLU). In supervised learning, training targets $y$ are known and a loss $\mathcal{L}_y(F_\theta(x))$ is minimized as a function of the network parameters $\theta$. A typical loss for a classification task is the cross entropy between the network output and the empirical target distribution, while the mean-squared error is commonly used for a regression task. The proposed method can be applied to constrain any linear activations $z^{(l)} = f_{\theta_l}^{(l)}(a^{(l-1)})$ or non-linear activations $a^{(l)} = \sigma(z^{(l)})$. In most cases, one would like to constrain the output $F_\theta(x)$.

The feasible set for $m$ linear inequality constraints in $d$ dimensions is the convex polyhedron

$$\mathcal{C} := \left\{ z \middle| Az \le b, A \in \mathbb{R}^{m \times d}, b \in \mathbb{R}^m \right\} \subseteq \mathbb{R}^d . \tag{2}$$

A suitable description of the convex polyhedron $\mathcal{C}$ is obtained by the decomposition theorem for polyhedra.

**Theorem 1** (Decomposition of polyhedra, Minkowski-Weyl). *A set $\mathcal{C} \subset \mathbb{R}^d$ is a convex polyhedron of the form* (2) *if and only if*

$$\mathcal{C} = \mathrm{conv}(v_1, \ldots, v_n) + \mathrm{cone}(r_1, \ldots, r_s)$$

$$= \left\{ \sum_{i=1}^n \lambda_i v_i + \sum_{j=1}^s \mu_j r_j \middle| \lambda_i, \mu_j \ge 0, \sum_{i=1}^n \lambda_i = 1 \right\} \tag{3}$$

*for finitely many vertices $\{v_1, \ldots, v_n\}$ and rays $\{r_1, \ldots, r_s\}$.*

*Furthermore, $\mathcal{C} = \left\{ z | Az \le 0, A \in \mathbb{R}^{m \times d} \right\}$ if and only if*

$$\mathcal{C} = \mathrm{cone}(r_1, \ldots, r_s) \tag{4}$$

*for finitely many rays $\{r_1, \ldots, r_s\}$.*

The theorem states that an intersection of half-spaces (half-space or H-representation) can be written as the Minkowski sum of a convex combination of the polyhedron's vertices and a conical combination of some rays (vertex or V-representation). One can switch algorithmically between these two viewpoints via the double description method (Motzkin et al., 1953; Fukuda & Prodon, 1996), which we discuss in the following. Thus, the H-representation, which is natural when modeling inequality constraints, can be transformed into the V-representation, which can be incorporated into gradient-based neural network training.

In this paper, we focus on *homogeneous* constraints of the form (4), for which the feasible set is a polyhedral cone. Due to the special structure of this set, we can avoid to work with the convex combination parameters in (3), which is numerically advantageous (Section 3.5), and we can efficiently combine modeling constraints and domain constraints, such as a $[-1, 1]$-pixel domain for images (Section 3.3). Such a polyhedral cone is shown in Figure 2.

### 3.1 Double description method

The double description method converts between the half-space and vertex representation of a system of linear inequalities. It was originally proposed by Motzkin et al. (1953) and further refined by Fukuda & Prodon (1996).[2] Here, we are only interested in the conversion from H-representation to V-representation for homogeneous constraints (4),

$$\mathcal{H} \to \mathrm{cone}(r_1, \ldots, r_s) . \tag{5}$$

---

[1] Formally, $\sigma$ maps between different spaces for different layers and may also be a different element-wise non-linearity for each layer. We omit such details in favor of notational simplicity.

[2] In our experiments we use `pycddlib`, which is a Python wrapper of Fukuda's `cddlib`.

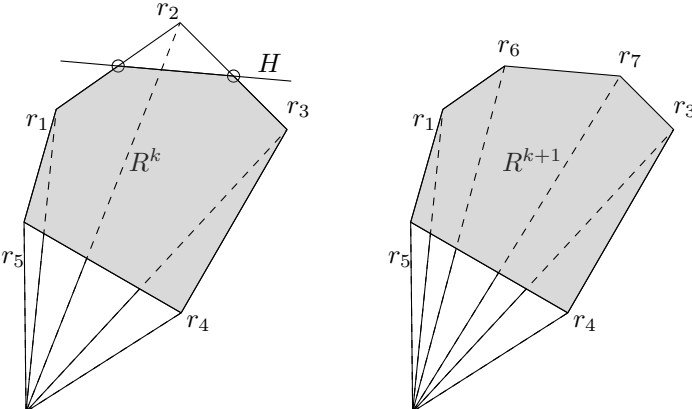

Figure 2: Diagram illustrating an iteration of the double description method. Adding a constraint to the $k$-constraint set $A^k$ at iteration $k+1$ introduces a hyperplane $H$. The intersection points of $H$ with the boundary of the current polyhedron $R^k$ (marked by $\circ$) are added as rays $r_6$ and $r_7$ to the polyhedral cone. The ray $r_2$ is cut-off by the hyperplane $H$ and is removed from $R^k$. The result is the next iterate $R^{k+1}$.

The core algorithm proceeds as follows. Let the rows of $A$ define a set of homogeneous inequalities and let $R = [r_1, \ldots, r_s]$ be the matrix whose columns are the rays of the corresponding cone. Here, $(A, R)$ form a double description pair. The algorithm iteratively builds a double description pair $(A^{k+1}, R^{k+1})$ from $(A^k, R^k)$ in the following manner. The rows in $A^k$ represent a $k$-subset of the rows of $A$ and thus define a convex polyhedron associated with $R^k$. Adding a single row to $A^k$ introduces an additional half-space constraint, which corresponds to a hyperplane. If the vector $r_i - r_j$ for two columns $r_i, r_j$ of $R^k$ intersects with this hyperplane then this intersection point is added to $R^k$. Existing rays that are cut-off by the additional hyperplane are removed from $R^k$. The result is the double description pair $(A^{k+1}, R^{k+1})$. This procedure is shown in Figure 2.

Adding a hyperplane might drastically increase the number of rays in intermediate representations, which, in turn, contribute combinatorially in the subsequent iteration. In fact, there exist worst case polyhedra for which the algorithm has exponential run time as a function of the number of inequalities and the input dimension, as well as the number of rays (Dyer, 1983; Bremner, 1999). Overall, one can expect the algorithm to be efficient only for problems with a reasonably small number $m$ of inequalities and dimension $d$.

## 3.2 INTEGRATION IN NEURAL NETWORK ARCHITECTURES

We parameterize the homogeneous form (4) via a neural network layer. This layer takes as input some (latent) representation of the data, which is mapped to activations satisfying the desired hard constraints. The algorithm is provided with the H-representation of linear inequality constraints, i.e., a matrix $A \in \mathbb{R}^{m \times d}$ for $m$ constraints in $d$ dimensions to specify the feasible set (4). At initialization, we convert this to the V-representation via the double description method (Section 3.1). This corresponds to computing the set of rays $\{r_1, \ldots, r_s\}$ to represent the polyhedral cone. During training, the neural network training algorithm is used to optimize within in the feasible set. There are two critical aspects in this procedure. First, as outlined in Section 3.1, the run-time complexity of the double description method may be prohibitive. Conceptually, the proposed approach allows for significant compute time at initialization to obtain an algorithm that is very efficient at training and test time. Second, we must ensure that the mapping from the latent representation to the parameters integrates well with the training algorithm. We assume that the model is trained with gradient-based backpropagation, as is common for current deep learning applications. The constraint layer comprises a batch normalization layer and an affine mapping (fully-connected layer with biases) followed by the element-wise absolute value function that ensures the non-negativity required by the conical combination parameters. In theory, any function $f : \mathbb{R} \to \mathbb{R}_{\geq 0}$ would fulfill this requirement; however, care must be taken to not interfere with backpropagated gradients.

### 3.3 Combining modeling and domain constraints

Domain constraints are often formulated as unit box constraints, $\mathcal{B} \coloneqq \{x \in \mathbb{R}^d | -1 \leq x_i \leq 1\}$, such as a pixel domain for images. Box constraints are particularly unfit to be converted using the double description method because the number of vertices is exponential in the dimension. Therefore, we distinguish *modeling constraints* and *domain constraints* and only convert the former into V-representation. Based on this representation, we obtain a point in the modeling constraint set, $x \in \mathcal{C}$. However, this point may not be in the unit box $\mathcal{B}$. To arrive at a point in the intersection $\mathcal{C} \cap \mathcal{B}$, we normalize $x$ by its infinity norm if $x \notin \mathcal{B}$, $\hat{x} = x / \max\{\|x\|_\infty, 1\}$. Indeed, $\hat{x} \in \mathcal{C} \cap \mathcal{B}$ since scaling by a positive constant remains in the cone, i.e., if $x \in \mathcal{C}$, then $\alpha x \in \mathcal{C} \; \forall \alpha \geq 0$.

### 3.4 Applications of homogeneous linear inequality constraints

A natural application of constraints of the form $Ax \leq 0$ is a parameterization of a set of binary classifiers. If each row $a_i$ of $A$ is such a binary classifier, then the method presented in this paper parameterizes the set $\{x | a_i^T x \leq 0 \; \forall i\}$. Consequently, it can be guaranteed that neural network activations satisfy a set of binary criteria. Another domain is to express certain direct relations between neural network activations. Notably, one can guarantee mathematical properties such as monotonicity via $x_{i+1} \geq x_i$ and convexity via $x_{i+1} - 2x_i + x_{i-1} \geq 0$.

### 3.5 Extension to general linear inequality constraints

The proposed method takes advantage of the special structure of a polyhedral cone to efficiently combine modeling and domain constraints (Section 3.3). General linear inequality constraints of the form $Ax \leq b$ without restrictions on $A$ and $b$ possibly require the conic and convex component of (3) for their V-representation. The main approach of this paper may be used in this case, i.e., our layer additionally needs to predict convex combination parameters. However, we observed slow convergence, which we ascribe to the simplex parameterization for the convex combination parameters. We used a softmax function $f(x)_i = \exp(x_i) / \sum_{j=1}^m \exp(x_j)$ to enforce the constraints $\lambda_i \geq 0, \sum_{i=1}^m \lambda_i = 1$ of the convex combination parameters in (3). This function has vanishing gradients when one $x_i$ is significantly greater than the other vector entries. Furthermore, this most general setting does not allow for efficient incorporation of domain constraints, as this would require an efficient parameterization of the intersection of a general convex polyhedron and the unit box.

## 4 Numerical Results

We compare the proposed *constraint parameterization* algorithm with an algorithm that trains without constraints, but requires a projection step at test time. We call this latter algorithm *test time projection*. We analyze these algorithms in two different settings. In an initial experiment, we learn the orthogonal projection onto a constraint set to demonstrate properties of these algorithms. Here, the result can be compared to the optimal solution of the convex optimization problem. In a second experiment, consistent with our motivation to constrain the output of generative models, we apply these algorithms to a variational autoencoder. Finally, we evaluate the running time of inference for these problems and show that the proposed algorithm is significantly more efficient compared to the test time projection method.

We used the MNIST dataset (LeCun et al.) for both experiments (59000 training, 1000 validation, and 10000 test samples). We chose PyTorch (Paszke et al., 2017) for our implementation[3] and all experiments were performed on a single Nvidia Titan X GPU. All networks were optimized with the Adam optimizer and we evaluated learning rates in the range $[10^{-5}, 10^{-3}]$. The initial learning rate was annealed by a factor of $1/2$ if progress on the validation loss stagnated for more than 5 epochs. We used OSQP (Stellato et al., 2017) as an efficient solver to compute orthogonal projections.

Both experiments were performed with a checkerboard constraint with 16 tiles, where neighboring tiles are constrained to be on average either below or above pixel domain midpoint. For a $[-1, 1]$-pixel domain, the tiles' average intensity is positive or negative, respectively. The initial computational cost of converting these constraints into V-representation via the double description

---

[3]Our implementation will be publicly available.

method is negligible (less than 1s). We observed that it is numerically advantageous to activate unit box scaling after the constraint parameterization model was initially optimized only with modeling constraints for a specified number of epochs.

One might consider OptNet (Amos & Kolter, 2017) and an analogous version of the method introduced by Pathak et al. (2015) as baselines. However, these approaches incur a significant drawback for the setup presented in this paper as they are are computationally expensive at training time. An OptNet layer solves a generic quadratic program as a differentiable network layer, which scales cubically with the number of variables and constraints. The method by Pathak et al. (2015) for the regression problems in this paper alternates between optimization steps in the network parameters via a variant of stochastic gradient descent and projecting the network output onto the constraints, which is computationally expensive.

## 4.1 ORTHOGONAL PROJECTION ONTO A CONSTRAINT SET

We learn an orthogonal projection to demonstrate general properties of both algorithms. For given linear inequalities specified in H-representation, we solve the following problem:

$$\min_{z \in \mathbb{R}^d} \|z - y\|_2 \quad \text{s.t. } Az \leq 0 \quad , \tag{6}$$

where $y$ is an MNIST image. Here, the problem is convex; therefore, the global optimum can be readily computed and compared to the performance of the learning algorithms. In this setting, we can expect that training an unconstrained network with subsequent projection onto the constraint set at test time yields good results, which can be seen as follows. Let $\mathcal{P}_\mathcal{C}(y) := \arg\min_{z \in \mathcal{C}} \|z - y\|_2$ be the orthogonal projection onto the constraint set $\mathcal{C}$ and denote the mean-squared error as $\mathcal{L}_y(x) := \|x - y\|_2$. Both mappings are Lipschitz continuous with Lipschitz constant $L = 1$. Consequently, for an output $\hat{y}$ of an unconstrained model,

$$\left| \mathcal{L}_y(\mathcal{P}_\mathcal{C}(\hat{y})) - \mathcal{L}_y(\mathcal{P}_\mathcal{C}(y)) \right| \leq \left\| \mathcal{P}_\mathcal{C}(\hat{y}) - \mathcal{P}_\mathcal{C}(y) \right\|_2 \leq \|\hat{y} - y\|_2 \ , \tag{7}$$

where, by definition, the term $\mathcal{L}_y(\mathcal{P}_\mathcal{C}(y))$ is the optimal value of problem (6). The training algorithm fits $\hat{y}$ to $y$; therefore, projecting the unconstrained output $\hat{y}$ onto the constraint set will yield an objective value that is close to the optimal value of the constrained optimization problem.

To have a comparable number of parameters for both methods, we use a single fully-connected layer in both cases. For the unconstrained model, we employ an $FC(784, 784)$ layer, and for the constrained model we employ an $FC(784, n_r)$ layer with $n_r = 1552$ many rays to represent the constraint set in V-representation. Additionally, the constraint layer first applies a batch normalization operation (Ioffe & Szegedy, 2015). Both models were optimized with an initial learning rate of $10^{-4}$, which was annealed by a factor of $0.1$ if progress on the validation loss stagnated for more than 5 epochs. The batch size was chosen to be 256. The unit box constraints were activated after 25 epochs. Additionally, the data for training the model with all constraints being active is shown. This mode eventually results in worse generalization. Figure 3 shows that the mean-squared validation objective for both algorithms converges close to the average optimum. The constraint parameterization method has a larger variance and optimality gap, which hints at the numerical difficulty of training the constrained network. To be precise, the best average validation error during training is within $9\%$ of the optimum for the constraint parameterization method and within $1\%$ of the optimum for the test time projection method. Figure 4 shows a test set sample and the respective output of the learned models.

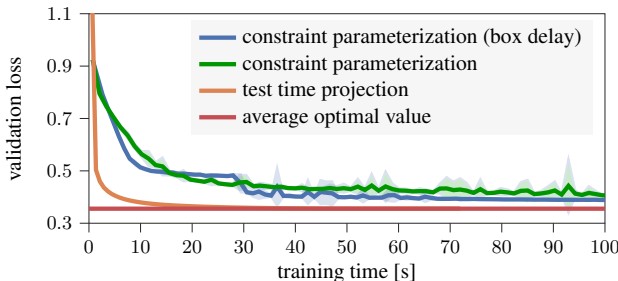

Figure 3: Mean-squared validation loss averaged over all pixels for 10 runs; shaded area denotes standard deviation. The objective function (6) is computed on a held-out validation set for the proposed constraint parameterization method and unconstrained optimization with subsequent test time projection. The average optimum over the validation set is obtained as a solution to a convex optimization problem. For the *box delay* curve, the box constraints are activated after 25 epochs (after $\sim 30$s), which results in better generalization. The best average validation error during training is within 9% of the optimum for the constraint parameterization method with box constraint delay and within 1% of the optimum for the test time projection method.

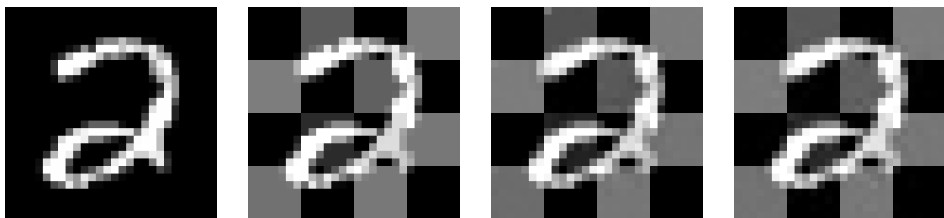

Figure 4: Learning to solve the orthogonal projection onto a constraint set as defined in (6). From left to right: MNIST sample from a test set, optimal projection by solving a quadratic program, constraint parameterization model inference, and test time projection model inference.

## 4.2 CONSTRAINED GENERATIVE MODELING WITH VARIATIONAL AUTOENCODERS

Variational autoencoders (VAE) are a class of generative models that are jointly trained to encode observations into latent variables via an encoder or inference network and decode observations from latent variables using a decoder or generative network (Kingma & Welling, 2014). We base our implementation on (Baumgärtner, 2018). The model has a fully-connected architecture:

$$\text{encoder: } FC(784, 256) - \text{ReLU} - FC(256, 2)$$
$$\text{decoder: } FC(2, 256) - \text{ReLU} - FC(256, 784) - \text{sigmoid} - \text{constraint}$$

Here, $\text{ReLU}(x) = \max(0, x)$ and the sigmoid non-linearity takes the form $\sigma(x) = 1/(1+\exp(-x))$. In contrast to a standard VAE, we constrain the samples generated by the model to obey a checkerboard constraint. The model was optimized with an initial learning rate of $10^{-4}$, which was annealed by a factor of $0.1$ if progress on the validation loss stagnated for more than 5 epochs. The batch size was chosen to be 64. The model was trained for 200 epochs while the unit box constraints were activated after 100 epochs. To generate images, we sample the latent space prior $z \sim \mathcal{N}(0, I)$ and evaluate the decoding neural network (Figure 5). The model is able to sample authentic digits while obeying the checkerboard constraint.

## 4.3 FAST INFERENCE WITH CONSTRAINED NEURAL NETWORKS

The main advantage of the proposed method over a simple projection method is a vast speed-up at test time. Since the constraint is incorporated into the neural network architecture, a forward pass has almost no overhead compared to an unconstrained network. On the other hand, for a network that was trained without constraints, a final projection step is necessary; this requires solving a convex

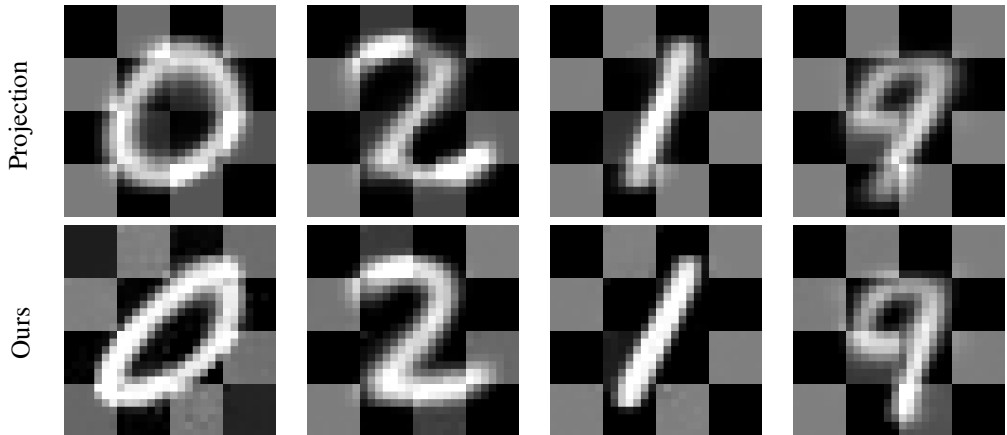

Figure 5: Samples from a constrained variational autoencoder trained with the test time projection method and our constraint parameterization method. The images represent authentic digits while satisfying the imposed checkerboard constraint. Inference is significantly faster using our method.

Table 1: Inference time for test time projection and constraint parameterization methods. Mean and standard deviation of running times are computed over 100 runs of 59000 samples with a batch size of 256.

| METHOD | PROJECTION | VAE |
|---|---|---|
| Test time projection | $82 \pm 1$ s | $40 \pm 1$ s |
| Constraint parameterization (ours) | $\mathbf{0.46 \pm 0.02}$ s | $\mathbf{0.75 \pm 0.04}$ s |

optimization problem, which is relatively costly. Table 1 shows inference times for both models for the above numerical experiments. The constraint parameterization approach is up to two orders of magnitude faster at test time compared to the test time projection algorithm.

## 5 CONCLUSION

To combine a data-driven task with modeling constraints, we have developed a method to impose homogeneous linear inequality constraints on neural network activations. At initialization, a suitable parameterization is computed and subsequently a standard variant of stochastic gradient descent is used to train the reparameterized network. In this way, we can efficiently guarantee that network activations – in the final or any intermediate layer – satisfy the constraints at any point during training. The main advantage of our method over simply projecting onto the feasible set after unconstrained training is a significant speed-up at test time of up to two orders of magnitude. An important application of the proposed method is generative modeling with prior assumptions. Therefore, we demonstrated experimentally that the proposed method can be used successfully to constrain the output of a variational autoencoder. Our method is implemented as a layer, which is simple to combine with existing and novel neural network architectures in modern deep learning frameworks and is therefore readily available in practice.

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
