# OpenReview forum: "Homogeneous Linear Inequality Constraints for Neural Network Activations"
_ICLR.cc/2020/Conference — Reject_

### Official Review · AnonReviewer1 · 2019-10-16
**Official Blind Review #1**

**Rating:** 1

**Review:**

The paper proposes a new faster algorithm to add inequality constraints to neural layers. The paper focuses on a novel constraining approach with seemingly superior scalability, and this is potentially a significant contribution. However the paper does not motivate the constraining at all. I am baffled by this, since one would assume at least some benefits from all of this work could be presented. The only mentions are binarization of the predictions (which softmax already does), and monotonicity/convexity of neurons, with no proposed benefits. The running example of the paper is the chessboard constraint, which is either pointless (fig1) or harmful (fig5). Without justification and motivation the method has no merit and won’t have any impact in the machine learning community.

The monotonicity constraint could have a huge impact for MCMC sampling of neural parameters since it can reduce away all multimodalities of the posterior caused by reordering nodes or layers.

I had hard time following the method, and I its not clear how the neural network is modified and how backpropagation is performed with the contraints. It is not defined properly how the constrained optimisation works. Apparently additional neural layers are added that map z's to r's. The backpropagation in the constrained case is undefined. Here an algorithm box or schematic figure comparing unconstrained and constrained NN architectures would be extremely helpful. It’s also not explained how are modelling/domain constraints different.

The paper does not compare to the earlier constrained methods (Marquaz-Neila or OptNet), and thus there is no demonstration of the methods claimed superior computational efficiency. The paper also does not make very clear the different constraining approach advantages and tradeoffs. A comparison table would be help a lot.

The method is interesting, novel and seemingly efficient; but it is insufficiently defined, the method is not motivated and experiments are quite weak with little comparisons and no experiments with practical value.


**Experience Assessment:**

I have read many papers in this area.

**Review Assessment: Checking Correctness Of Derivations And Theory:**

I assessed the sensibility of the derivations and theory.

**Review Assessment: Checking Correctness Of Experiments:**

I assessed the sensibility of the experiments.

**Review Assessment: Thoroughness In Paper Reading:**

I read the paper at least twice and used my best judgement in assessing the paper.

---

> ### Author Response · Authors · 2019-11-14
> **Reply to Review #1**
>
> Thanks for pointing out the MCMC example.
> We agree that applications of homogeneous linear inequality constraints are limited, but think that the paper has merit in presenting novel ideas of how to reparameterize a neural network to incorporate such constraints.
>
> Let us clarify how the constraints are incorporated as a layer. The feasible set for homogeneous linear inequality constraints is a polyhedral cone; we compute the rays $r_i$ of the cone prior to training. Now, every point in the feasible set can be written as $\sum_i \mu_i r_i$, where $\mu_i \geq 0$. We parameterize these $\mu_i$, e.g., through a linear layer followed by the absolute value function $\mu = |Wa + b|$. Here, the $a$ are some latent activations of the network. Such a system can now be trained end-to-end and the output is guaranteed to be in the feasible set thanks to this reparameterization.
>
> As pointed out in our paper, a comparison with the methods you have mentioned is futile. The scaling behavior of these methods is discussed in the respective sources. Once the V-representation is computed prior to training, the subsequent training and inference phases have no significant overhead. Consequently, a comparison with methods that solve sub-optimization problems at training time is not meaningful here.

---

### Official Review · AnonReviewer3 · 2019-10-18
**Official Blind Review #3**

**Rating:** 3

**Review:**

The paper presents a method for imposing linear inequality in neural networks. Although the contribution of the paper is potentially significant, some details are not clearly described.

In the proposed method, the inequality constraints are converted from the representation with a matrix to the representation with rays that represents the cone which satisfies the constraints. The neural network is trained so as to satisfy the constraint represented by rays. However, I do not understand how to train the neural network to satisfy the constraint represented by rays, which is actually the core of the algorithm. I cannot see how satisfaction of constraints are guaranteed from the current description.

The empirical results show that the variational autoencoder trained with the proposed method can generate images that satisfy linear constraints. However, the evaluation is limited to a checkerboad constraint, and other examples of practical linear constraints are not clear.

Due to the unclear algorithm description and limited empirical results, I give weak reject to the paper. However, I'm happy raise the score if authors clarify some points in the rebuttal.

I would like authors to address the following points in the rebuttal:

- I do not understand the procedure of the proposed method. Especially, I do not understand how to achieve this part: "During training, the neural network training algorithm is used to optimize within in the feasible set."
Please elaborate it in the rebuttal. Please describe how the satisfaction of the constraints are guaranteed.

- I recommend authors to put a pseudo-code of the proposed algorithm for clarity.

- In Section 4.1, it is stated that the constraint layer is added to the neural network. However, the computation performed in the constraint layer is not clear. Please describe it.

- The checker board constraint on MNIST images is interesting, but it would be better to show more examples of linear constraints. If possible, please give some more examples of linear constraints and their results.


**Experience Assessment:**

I do not know much about this area.

**Review Assessment: Checking Correctness Of Derivations And Theory:**

N/A

**Review Assessment: Checking Correctness Of Experiments:**

I assessed the sensibility of the experiments.

**Review Assessment: Thoroughness In Paper Reading:**

I read the paper at least twice and used my best judgement in assessing the paper.

---

> ### Author Response · Authors · 2019-11-14
> **Reply to Review #3**
>
> Let us clarify how the constraints are incorporated as a layer. The feasible set for homogeneous linear inequality constraints is a polyhedral cone; we compute the rays $r_i$ of the cone prior to training. Now, every point in the feasible set can be written as $\sum_i \mu_i r_i$, where $\mu_i \geq 0$. We parameterize these $\mu_i$, e.g., through a linear layer followed by the absolute value function $\mu = |Wa + b|$. Here, the $a$ are some latent activations of the network. Such a system can now be trained end-to-end and the output is guaranteed to be in the feasible set thanks to this reparameterization.

---

### Official Review · AnonReviewer5 · 2019-10-30
**Official Blind Review #5**

**Rating:** 3

**Review:**

This paper proposes a method to impose linear inequality constraints on neural network activations. The method is implemented at initialization (by converting the H-representation to the V-representation) and during training (by modifying the network architecture). Experiments on two setups (projection and VAE+projection on a checkerboard pattern on MNIST) demonstrate a 2-orders of magnitude speed-up with respect to test-time projection (computed with OSQP).

The contributions claimed are:
* Novel technique to impose inequality constraints on neural network activations.
* Significant speed-up w.r.t. other techniques (at test time).

Overall, the approach is well motivated, and well-placed in the literature. However, the experimental analysis does not support all the claims made by the authors as it focuses on a single dataset (i.e., MNIST) and single constraint (i.e., checkerboard pattern). Additionally, while the authors argue that there is no manual trade-off between constraint satisfaction and data representation, the experiment in Fig. 3 appears to show that there is some manual trade-off (using box delay). As such, I am currently inclined to give a "weak reject" score.

The method is clear and the idea of using softmax to get a convex combination of the vertices of the V-representation to guarantee constraint satisfaction is reasonable.

1) The softmax used to satisfy the constraints is preceded by a batch normalization layer. I would expect batch normalization to interfere with the ability to saturate the softmax activation and thus prevent the network from reaching optimality. Could the authors provide experiments that justify the use of the batch normalization layer?

2) An important factor that is unexplored in this manuscript is the softmax temperature. A good scheduling of that temperature could help the optimization. Have authors tried different temperature values?

3) The use of softmax and the integration of the constraints as a network layer seem to create some difficulty during training (even with the rather simple checkerboard pattern used in the experiment). The loss appears to reach some plateau (9% from optimal) and, thus, there appears to be some trade-off between reconstruction and projection. The authors should provide more experiments to explain that trade-off. That trade-off is also visible on Fig. 5 (the zero has a significantly different shape).

4) The method is solely compared to test time projection. Could the authors implement other techniques (if feasible)? e.g., OptNet. Overall, it would helpful to add more setups and different types of constraints (other than a last-layer projection; e.g., monotonicity).

**Experience Assessment:**

I have published one or two papers in this area.

**Review Assessment: Checking Correctness Of Derivations And Theory:**

N/A

**Review Assessment: Checking Correctness Of Experiments:**

I carefully checked the experiments.

**Review Assessment: Thoroughness In Paper Reading:**

I read the paper at least twice and used my best judgement in assessing the paper.

---

> ### Author Response · Authors · 2019-11-14
> **Reply to Review #5**
>
> We agree that generally speaking a comparison with more methods is desirable. However, as pointed out in the paper, the methods mentioned in the related work section don’t scale well and hence a comparison with our method is futile. Once the V-representation is computed prior to training, the subsequent training and inference phases have no significant overhead, whereas the other methods solve a sub-optimization problem at training time.
>
> While the softmax is indeed a sensible choice for general linear inequalities, the homogeneous case presented in this paper in fact does not use a softmax. Since the feasible set for homogeneous linear inequalities is a polyhedral cone, all we need is a function that maps to positive values, which we can then use as conic combination parameters. We use the absolute value function for that purpose.
>
> You are right about pointing out that there is an implicit trade-off between constraint satisfaction and data representation in the box delay experiments (Sec. 4.1). We will remove the statement in a final version.

---

### Author Response · Authors · 2019-11-14
**Replies to Reviewer Feedback**

We thank the reviewers for their time and constructive feedback. We have posted individual comments below each review.

---

### Decision · Program_Chairs · 2019-12-19

**Decision:**

Reject

**Comment:**

The authors propose a framework for incorporating homogeneous linear inequality constraints on neural network activations into neural network architectures. The authors show that this enables training neural networks that are guaranteed to satisfy non-trivial constraints on the neurons in a manner that is significantly more scalable than prior work, and demonstrate this experimentally on a generative modelling task.

The problem considered in the paper is certainly significant (training neural networks that are guaranteed to satisfy constraints arises in many applications) and the authors make some interesting contributions. However, the reviewers found the following issues that make it difficult to accept the paper in its present form:
1) The setting of homogeneous linear equality constraints is not well-motivated and the significance of being able to impose such constraints is not clearly articulated in the paper. The authors would do well to prepare a future revision documenting use-cases motivated by practical applications and add these to the paper.
2) The experimental evaluation is not sufficiently thorough: the authors evaluate their method on an artificial constraint involving a "checkerboard pattern" on MNIST. Even in this case, the training method proposed by the authors seems to suffer from some issues, and more thorough experiments need to be conducted to confirm that the training method can perform well across a variety of datasets and constraints.

Given these issues, I recommend rejection. However, I encourage the authors to revise their work on this important topic and prepare a future version including practical examples of the constraints and experiments on a variety of prediction tasks.